# Gender Influences Gut Microbiota among Patients with Irritable Bowel Syndrome

**DOI:** 10.3390/ijms241310424

**Published:** 2023-06-21

**Authors:** Paulina Pecyna, Marcin Gabryel, Dorota Mankowska-Wierzbicka, Dorota M. Nowak-Malczewska, Katarzyna Jaskiewicz, Marcelina M. Jaworska, Hanna Tomczak, Malgorzata Rydzanicz, Rafal Ploski, Marian Grzymislawski, Agnieszka Dobrowolska, Marzena Gajecka

**Affiliations:** 1Chair and Department of Genetics and Pharmaceutical Microbiology, Poznan University of Medical Sciences, 60-806 Poznan, Poland; pa.pecyna@ump.edu.pl (P.P.); dmnowak@ump.edu.pl (D.M.N.-M.); marcelinajaworska@ump.edu.pl (M.M.J.); hannatomczak@interia.pl (H.T.); 2Department of Gastroenterology, Dietetics and Internal Diseases, Poznan University of Medical Sciences, 60-355 Poznan, Poland; mgabryel@ump.edu.pl (M.G.); dorotamw@ump.edu.pl (D.M.-W.); mariangrzym@ump.edu.pl (M.G.); agdob@ump.edu.pl (A.D.); 3Institute of Human Genetics, Polish Academy of Sciences, 60-479 Poznan, Poland; jaskiewiczkatarzynaa@gmail.com; 4Central Microbiology Laboratory, H. Swiecicki Clinical Hospital at the Poznan University of Medical Sciences, 60-355 Poznan, Poland; 5Department of Medical Genetics, Medical University of Warsaw, 02-106 Warsaw, Poland; malgorzata.rydzanicz@wum.edu.pl (M.R.); rafal.ploski@wum.edu.pl (R.P.)

**Keywords:** microbiota composition alterations in IBS, 16S rRNA sequencing, gender influence, gut microbiota

## Abstract

Irritable bowel syndrome (IBS) is a chronic functional gastrointestinal disease that affects approximately 11% of the general population. The gut microbiota, among other known factors, plays a substantial role in its pathogenesis. The study aimed to characterize the gut microbiota differences between patients with IBS and unaffected individuals, taking into account the gender aspect of the patients and the types of IBS determined on the basis of the Rome IV Criteria, the IBS-C, IBS-D, IBS-M, and IBS-U. In total, 121 patients with IBS and 70 unaffected individuals participated in the study; the derived stool samples were subjected to 16S rRNA amplicon sequencing. The gut microbiota of patients with IBS was found to be more diverse in comparison to unaffected individuals, and the differences were observed primarily among *Clostridiales*, *Mogibacteriaceae*, *Synergistaceae*, *Coriobacteriaceae*, *Blautia* spp., and *Shuttleworthia* spp., depending on the study subgroup and patient gender. There was higher differentiation of females’ gut microbiota compared to males, regardless of the disease status. No correlation between the composition of the gut microbiota and the type of IBS was found. Patients with IBS were characterized by more diverse gut microbiota compared to unaffected individuals. The gender criterion should be considered in the characterization of the gut microbiota. The type of IBS did not determine the identified differences in gut microbiota.

## 1. Introduction

The human body is characterized by many microbiomes depending on its various locations/regions, such as the oral cavity, respiratory tract, vagina, and intestines [1]. The microbiota inhabiting the gut undergoes modifications throughout human life, in response to endogenous and exogenous factors [2,3,4]. Irritable bowel syndrome (IBS) is a chronic functional gastrointestinal disease that affects approximately 11% of people worldwide [5]. It is estimated that IBS is the most common disease in terms of visits to gastroenterologists and the deteriorating quality of life for patients [5]. Currently, IBS diagnosis is based on the Rome IV criteria [6]. Patients with IBS often suffer from bloating, abdominal pain or discomfort, distension, and stool alterations [7].

The relationship between IBS and small intestinal bacterial overgrowth (SIBO) in patients (in a range of 4% to 78%) [8] has been reported, and the investigated selected bacteria were found to be more abundant in the gut microbiome [9,10], collectively pointing to a substantial role of microbiota in IBS.

The goal of this study was to characterize the bacterial element of the gut microbiome in patients with IBS compared to non-IBS control group individuals.

Patients and non-IBS control group were divided into subgroups of females and males to verify the previously evaluated influence of gender on the composition of the gut microbiome [11]. As there are IBS-C (constipation), IBS-D (diarrhea), IBS-M (mixed), and IBS-U (undefined), types of the IBS [6], we evaluated the microbiota in stool samples taking this division into account.

## 2. Results

In total, 121 patients with IBS (70 females and 51 males) and 70 non-IBS control group individuals without gastrointestinal complaints (40 female and 30 male individuals) were ascertained.

The clinical characteristics of patients with IBS and the non-IBS control group, as well as the subgroups of patients with particular types of IBS (the IBS-C IBS-D, IBS-M, and IBS-U) were compiled in Table 1. Patients’ answers regarding, e.g., abdominal pain, stool form, or frequency of defecation among patients with IBS are compiled in Table 2.

In total, 121 and 70 stool samples were collected from the patients with IBS and non-IBS control group, respectively. After the 16S rRNA amplicon sequencing, the sequencing data derived from 190 DNA samples had adequate quality to proceed further.

A total of 5568 ASVs (amplicon sequence variants) were identified, which occurred 1,660,070 times in all samples. The mean frequency of ASVs in the samples was 8691, while the median was 7474.

The assessment of the microbiota was carried out based on the relative abundance at seven taxonomic levels. In total, 10 phyla, 13 classes, 15 orders, 19 families, and 23 genera of microorganisms were identified, as indicated in Figure 1 and Appendix A. At the phylum level, our analysis revealed differences in the abundance of microbial taxa between non-IBS control individuals and IBS patients. Specifically, we observed higher abundances of *Firmicutes*, *Bacteroidetes*, *Proteobacteria*, and *Tenericutes* in stool samples derived from non-IBS individuals compared to IBS patients. Conversely, the abundances of *Actinobacteria*, *Verrucomicrobia*, *Cyanobacteria*, *Synergistetes*, and *Fusobacteria* phyla were found to be higher among IBS patients compared to non-IBS individuals. Furthermore, we conducted an analysis of the *Firmicutes*/*Bacteroidetes* ratio (F/B ratio) in both study groups, revealing a higher ratio among patients with IBS (Figure 2).

Based on the results of the ANCOM analysis, differences between patients with IBS and individuals without symptoms among *Coriobacteriaceae* (W = 40) and *Clostridiales* (W = 30) ASVs at the family level were identified. In addition, the analysis pointed to differences in *Coriobacteriaceae* (W = 66), *Clostridiales* (W = 35), and *Candidatus Phytoplasma* (W = 6) at the genus level (Appendix A).

Assessment of α-diversity revealed the differences between the gut microbiota of patients with IBS and unaffected individuals (*p* < 0.0001) (Appendix A).

Using Jaccard and Bray–Curtis dissimilarity tests, it was found that the gut microbiota composition of patients with IBS was more diverse considering the microbiota composition compared to individuals without gastrointestinal symptoms i (Figure 3). The use of weight-UniFrac and unweight-UniFrac tests indicated that the phylogenetic origin of the gut microbiota in both studied groups was similar (Figure 4).

Comparing the gut microbiota of females with IBS (without taking into account the types of IBS) and female controls, the relative abundance of bacteria from *Clostridiales* (*p* = 0.019) order, *Mogibacteriaceae* (*p* = 0.044) and *Synergistaceae* (*p* = 0.036) families, and *Blautia* spp. (*p* = 0.040) was found to be increased among females with IBS. In contrast, *Shuttleworthia* spp. (*p* < 0.001) was overrepresented in female controls. The detailed data are presented in the Appendix A.

Regarding the differences between males with IBS and male controls, the abundance of *Coriobacteriaceae* increased among males with IBS (*p* < 0.001). In contrast, the abundance of *Shuttleworthia* spp. was reduced (*p* < 0.001) (Appendix A). The results concerning order, class, family, and genus taxonomic levels are presented in Figure 2.

Comparing α-diversity scores, differences in the microbiota abundance between females with IBS and female controls (*p* < 0.0001), as well as males with IBS and male controls (*p* = 0.028), were found. Results are presented in Appendix A.

### 2.1. There Are Differences in the Gut Microbiota between Females and Males with IBS

Assessing gender-specific microbiota differences in patients with IBS, we found that gene sequences of the bacteria order *Clostridiales* (*p* = 0.011) and family *Synergistaceae* (*p* = 0.018) were more abundant in samples obtained from females compared to males. Detailed data from the analysis are presented in Appendix A.

When evaluating α-diversity, differences in the abundance of the gut microbiota were found between females and males with IBS (*p* = 0.020) and females with IBS and control males (*p* < 0.0001).

### 2.2. The type of IBS Does Not Influence Microbiota Diversity

Regarding the influence of the IBS types on the gut microbiota, no differences (*p* > 0.05) were found when analyzing *Clostridiales* and *Synergistaceae*, which were indicated as significantly related to previously observed differences (Appendix A).

### 2.3. The Presence of Clostridiales, Mogibacteriaceae, Coriobacteriaceae, Synergistaceae, Shuttleworthia spp., and Blautia spp. Had No Effect on Abdominal Pain in Patients with IBS

Analysis of abdominal pain responses over the last three months (as per Table 1) and the relative abundance of *Clostridiales*, *Mogibacteriaceae*, *Coriobacteriaceae*, *Synergistaceae*, *Shuttleworthia* spp., and *Blautia* spp. revealed no effect of gut microbiota composition on abdominal pain. Detailed data are compiled in Appendix A.

## 3. Discussion

IBS is often diagnosed in patients who have undergone gastroenteritis in the past [12]. At the same time, other studies indicated that intestinal dysbiosis is the cause of the disease, resulting from abnormalities in the quantitative and qualitative composition of the gut microbiota [13,14,15], which were previously assessed based on data obtained using 16S rRNA gene phylogenetic microarray analysis [16] or real-time PCR techniques [17]. It is still not known whether intestinal diseases result directly from the dysbiosis of the intestinal microflora or are a consequence of other factors that can affect the intestinal microflora.

The published research results emphasize the significant influence of gut microbiota on human health [18,19]. Microbiota diversity may become an important ‘indicator’ when assessing microbiomes in the context of disease etiology. Less differentiation of the gut microbiota has been identified in acute and chronic diseases, including IBD [20]. Reduced diversity promotes the overgrowth of some microbes, as observed in patients with IBS [18,19]. Still, most of the gut microbiota has not been characterized in terms of its influence on human health.

Previously, *Clostridiales* were identified as more abundant in samples obtained from patients with IBS compared to samples of unaffected participants, applying the 16S rRNA [21]. Similarly, the use of the pyrosequencing technique in studies involving 37 patients with IBS and 20 individuals without gastrointestinal signs allowed for the identification of the more frequent presence of *Clostridiales* in the IBS group (*p* < 0.001) [22], which contradicts our study results, in which we observed no differences between patients with IBS and unaffected individuals.

The family of *Coriobacteriaceae* has an unclear role in IBS but influences the metabolism of bile acids or the hormonal balance (aldosterone), and its number in the gut is determined, among others, by physical activity [23,24]. Considering the positive effect of the presence of the bacteria in the gut on the status of human health, it was startling that *Coriobacteriaceae* was found to be more abundant in the gut microbiome of patients with IBS (*n* = 37) compared to control individuals (*p* < 0.01) [25]. Similarly, in our study, *Coriobacteriaceae* and *Clostridiales* were found to be more abundant among patients with IBS.

On the other hand, Chung et al., based on their research on stool samples and small gut biopsies obtained from 28 patients with IBS and 19 healthy individuals, have not reported significant differences in the presence of *Coriobacteriaceae* bacteria between the assessed biological materials the studied groups [26].

The presence of bacteria from the *Mogibacteriaceae* family was found to be significant in the gut microbiota, especially in the period up to the third year of life, regarding the occurrence of food allergies [27]. Moreover, these bacteria are more common in the elderly [28]; however, their status in IBS has yet to be clarified. In our study, *Mogibacteriaceae* was more abundant in stool samples obtained from patients with IBS compared to unaffected individuals.

*Synergistaceae* is one of the most common bacteria in the human gut microbiota [29]. These bacteria are known to be abundant in the periodontal environment [10]. However, available studies do not describe this microbial family in detail, referring to IBS patients and their importance in the IBS pathogenesis. It was observed that *Synergistaceae* bacteria were predominant in the gut microbiota of individuals with IBS, which was confirmed in our study too.

*Blautia* spp. comprises microorganisms with potentially probiotic properties [30]. In contrast, the influence of the *Blautia* genus in the gut on the early occurrence of breast cancer has been suggested, pointing to its involvement in cancerogenesis [31]. Interestingly, in a study of a group of patients with IBS (*n* = 62) and control individuals (*n* = 46), the gut microbiota of patients with IBS was characterized by a lower relative abundance of *Blautia* spp. [16]. The mentioned reports were contradicted by Liu et al., as they demonstrated that *Blautia* spp. were more abundant in the control gut microbiome [32]. We observed a higher relative abundance of *Blautia* spp. among females with IBS, in contrast to females without symptoms. Hence, the influence of *Blautia* spp. on human health is not established regarding the species level [30]. This aspect is important when drawing research conclusions because the impact of *Blautia* spp. on human health may result from differences at the species/strain level, which determines whether it has a positive or negative role.

*Anaerovorax* spp. has been reported to influence body weight regulation [33] and is one of the butyrate-producing gut microbes, produced through microbial fermentation of dietary fibers in the lower intestinal tract, which is essential for the appropriate gut functioning [34]. In our study, *Anaerovorax* spp. was found to be more abundant in samples derived from patients with IBS. This is in line with previous results of study based on stool samples of Finland patients, performed using the 16S rRNA gene phylogenetic microarray analysis with HITChip, in which a higher abundance of *Anaerovorax* spp. in the gut microbiota of IBS patients was recognized, compared to control individuals [16].

Numerous reports repeatedly emphasize the positive influence of butyrate producers on gut health [35]. There is meager information about the relationships between butyrate producers, including *Shuttleworthia* spp. and the remaining gut microbiota [36]. The role of *Shuttleworthia* spp. in IBS pathogenesis is still unexplained. However, it has been reported, based on the results of stool samples obtained using 16S rRNA sequencing (V3–V5), that *Shuttleworthia* spp. was more frequently observed in individuals with chronic constipation compared to controls (*p* < 0.05) [37]. In our research, *Shuttleworthia* spp. was more frequently identified in non-IBS control individuals compared to patients with IBS, and the differences in abundances of *Coriobacteriaceae, Clostridiales,* and *Candidatus Phytoplasma* between patients with IBS and unaffected individuals were found in ANCOM analysis.

Overall, in our study, patients with IBS were found to have more heterogeneous gut microbiota than those without gastrointestinal symptoms. It has been previously observed that gut microbiota was more diverse (*p* = 0.013) among patients with IBS (*n* = 80), comparing to non-IBS control individuals (*n* = 21) [38]. Furthermore, in another study, the gut microbiomes of patients with IBS (*n* = 29) were characterized as more diverse compared to the unaffected participants (*n* = 23) [39].

In contrast, Tap et al. reported no differences between the microbiota gut composition of patients with IBS (*n* = 110) and individuals without gastrointestinal complaints (*n* = 39), regarding the types of IBS [9], as well as Barandouzi et al., who did not notice differences between studied groups (IBS, *n* = 80; control group, *n* = 21) [38]. Moreover, Pozuelo et al. reported that IBS gut microbiota is characterized by less diversity compared to control individuals (*p* < 0.01) [40], while Hugerth et al. showed that the composition diversity of gut microbiota among patients with IBS and control individuals was similar (Bray–Curtis test, *p* = 0.096) [41].

### 3.1. Differences Regarding Gender

In the majority of studies on IBS, the criterion of gender has not been taken into account. Therefore, we assessed the influence of gender on the bacterial component in patients with IBS and non-IBS control individuals.

It has been proven that diseases of the digestive system are more common among females compared to males. It has also been shown that females are much more likely to suffer from IBS (2–2.5:1) [42,43] or chronic diarrhea of unknown etiology [44]. These data suggested that gender may influence microbiota research outcomes, and we confirmed that assumption.

Here, we revealed a difference in the presence of *Coriobacteriaceae* bacteria in the gut microbiota between males and females with their enrichment in male participants (*p* = 0.034). In contrast, in an animal research model, female mice exposed to 17ß-estradiol had a higher abundance of these bacteria in gut microbiota [45], which may suggest an effect of hormones on the gut bacteria. In the interpretation of the results, however, one should not forget about the discrepancies that occur between the human body and an animal experimental model.

Moreover, we observed that *Bacteroidales* were more frequently identified in females, independently of disease status. Previously, the gut microbiota has also been characterized by more frequent identification of bacteria from the same order, e.g., *Bacteroidaceae* and *Bacteroides* spp. [11].

In our study, we noticed differences between females and males regarding *Christensenellaceae* and *Christensenella* spp.; both bacteria were more frequently observed among females in comparison to males, skipping the study group. Interestingly, it has been found that *Christensenellaceae* is underrepresented in the gut microbiome of obese patients [46], but overrepresented in long-lived people [47]. However, differences between genders were not observed.

Furthermore, we found that the gut microbiota of females is characterized by a more frequent presence of *Anaerovorax* spp., *Mogibacterium* spp., and *Psuedobutyrivibrio* spp. in relation to the group of males, both with IBS and without gastrointestinal symptoms. Moreover, regardless of belonging to the study group, *Shuttleworthia* spp. was more often present in the gut of males, in contrast to female gut microbiota. All these findings underline the importance of considering gender aspects during microbiota assessments. 

Previously, differences in the gut microbiome were observed, taking into account the types of IBS, between control individuals (*n* = 16), IBS (IBS-U (*n* = 13), IBS-C (*n* = 9), and IBS-D (*n* = 13)), based on OTU values [48]. Moreover, the differences between the gut microbiome of patients with IBS-C and IBS-D (Tukey post-hoc, *p* = 0.011), as well as between IBS-C and IBS-M (*p* = 0.011), have been previously reported too [10]. In comparison, another study confirmed that the gut microbiota of patients with IBS-D and control individuals did not differ in terms of diversity based on Shannon’s indexes [49].

However, several studies have confirmed a higher number of *Clostridiales* and *Coriobacteriaceae* in patients with type IBS-C [9,10,50]. In our study, we conducted an analysis focusing on the specific microbiota that was previously identified as different in study groups, e.g., *Clostridiales* and *Synergistaceae*. Regarding IBS types, differences were not observed.

We did not observe any significant differences in the abundance of these taxa among the study groups. In contrast to the published findings, our results did not reveal any distinct microbial signatures associated with each IBS subtype.

In contrast to published data, we did not notice any distinguishing features of each type of IBS.

### 3.2. The Gut Microbiota Did Not Influence on Abdominal Pain Complaining by Patients with IBS

It has been established that intricate interactions between the gut microbiome and the brain play a significant role in modulating responses to visceral pain [51], particularly in relation to gastrointestinal disorders. However, the specific bacteria responsible for the sensation of pain have not yet been fully elucidated [52].

In our study, in the last three months prior to sampling, 48% of women suffered from abdominal pain daily, while 48% of men experienced abdominal pain more than once a week. This fact emphasizes the role of abdominal pain in IBS patients’ lives.

We did not observe a relationship between the several types of microbiota assessed and abdominal pain experienced by patients with IBS in the past last three months. In contrast to our findings, Brunkwall et al. reported that the presence and severity of bowel symptoms were associated with an increased abundance of *Fusobacterium* spp. (*p* = 0.02) [53]. Interestingly, the same authors showed that a higher abundance of *Blautia* spp. in gut microbiota was associated with diarrhea (*p* = 0.01), IBS (*p* = 0.002), and bowel symptoms (*p* = 0.0003) [53]. It was found in our study that *Blautia* spp. could influence IBS status, regarding the significant abundance of gut microbiota among females with IBS compared to females without gastrointestinal symptoms (*p* = 0.040). However, we did not find any correlation between patient complaints of abdominal pain and the abundance of *Clostridiales*, *Mogibacteriaceae*, *Coriobacteriaceae*, *Synergistaceae*, and *Shuttleworthia* spp.

## 4. Materials and Methods

Ascertainment, sample collection, and characteristics of patients with IBS and non-IBS control group.

Patients with IBS and control individuals aged 18 to 70 were recruited for the study. Patients with IBS underwent a medical examination performed by a gastroenterologist in the Department of Gastroenterology, Dietetics, and Internal Diseases, at Poznan University of Medical Sciences, Poznan, Poland (PUMS). They were qualified for the study if compliance with the Rome IV Criteria was found [54]. Then, patients with IBS were classified into subgroups, depending on their clinical characteristics: IBS-C, IBS-D, IBS-M and IBS-U), according to the Rome IV Criteria [6].

Control individuals were recruited after a medical examination performed by a gastroenterologist from the Department of Gastroenterology, Dietetics, and Internal Diseases at PUMS. The absence of gastrointestinal symptoms in the last three months, no elimination diet, and no metabolic disease were the criteria for inclusion of the person in the control group. The exclusion criteria for both study groups were smoking, pregnancy, metabolic diseases (including diabetes), hypertension, previous history of abdominal surgery, and the intake of antibiotics, antiviral and antifungal drugs, proton pump inhibitors, probiotics, prebiotics, symbiotics, herbs, and other supplements improving gut peristalsis, steroids, spasmolytic drugs, or antidepressants in the last three months. If a given person met at least one exclusion criterion, he/she did not qualify for the study.

As a biological material for the experiments, a stool sample was collected once from each study participant.

A questionnaire that included questions about abdominal/lower abdominal pain, stool consistency, frequency of bowel movements (considering pain), pain frequency, or general well-being was completed by each study participant in the presence of the gastroenterologist performing the clinical evaluation.

### 4.1. Microbial DNA Extraction and 16S Amplicon Sequencing

The genomic DNA from the stool samples was extracted using a ZymoBIOMICS DNA Miniprep Kit (Zymo Research, Irvine, CA, USA) according to the manufacturer’s protocol. The quality and quantity of the DNA samples were evaluated by Implen NanoPhotometr N60 UV-Vis spectrophotometry (Implen NanoPhotometer, Los Angeles, CA, USA).

Bacterial DNA sequences were evaluated using amplicon sequencing with universal primers 515F and 806R, targeting the V3-V4 hypervariable regions of the 16S rRNA gene. The library preparation was performed according to the 16S Metagenomic Sequencing Library Preparation—Preparing 16S Ribosomal RNA Gene Amplicons for the Illumina MiSeq System protocol in Illumina technology [55]. The paired-end sequencing (2 × 300 bp) was performed using the MiSeq platform (Illumina, San Diego, CA, USA) at the Medical University of Warsaw, Warsaw, Poland. For each sequencing run, each 16S rRNA amplicon pool was spiked-in with 10% of the reference PhiX Control v3 Library (Illumina) for improvement of the overall run quality. The sequencing run was performed with 10% of the reference PhiX Control v3 Library (Illumina) spike-in to improve the sequencing quality of 16S rRNA amplicon low-diversity libraries.

### 4.2. Data Analysis

#### Sequencing Data Workflow

Bioinformatic analysis of the raw 16S rRNA sequencing reads obtained was performed by the bioinformatics company, ideas4biology Sp. z o.o. in Poznan, Poland, in accordance with the protocol described elsewhere [56,57,58,59,60,61]. Briefly, initial reports of the sequencing read quality were generated for each sample separately using FastQC and aggregated using MultiQC [56]. Data obtained from the 16S rRNA sequencing were analyzed using QIIME 2 version 2019.7 [57]. The readings were verified based on the q2-dada2 function implementing the DADA2 algorithm. In order to improve the quality of the readings, artifacts, including PhiX sequences and chimeras, were removed using the indicated algorithm [58]. ASVs (amplicon sequence variants) of suitable quality and sequence length were obtained. Then, for the obtained ASVs, sequence alignment was performed using the MAFFT algorithm [59], implemented as part of the q2-alignment function [61]. ASVs were assigned to a given taxonomic level using a trained, naive Bayesian classifier based on the assumption of mutual independence of two variables. The above classifier was applied using the q2-feature classifier plugin to improve the parameters for optimizing the classifier’s performance [60]. To teach the classifier, the taxonomic Greengenes (16S) database was used (in version 13_8 99% OTUs reference sequences) [62], downloaded from ftp://greengenes.microbio.me/greengenes_release/gg_13_5/gg_13_8_otus.tar.gz (accessed on 30 September 2019).

### 4.3. Statistical Analyzes

Statistical analyses were performed using Statistica, version 13.4 (Dell Inc., Round Rock, TX, USA). The compliance of the empirical data distributions with the normal distribution was verified with the Shapiro–Wilk W test. Then, due to the inconsistency of the variables with a normal distribution, the Mann–Whitney U (for two groups) and Kruskal–Wallis (for three or more variables) tests were applied.

In addition, ANCOM (Analysis of Compositions of Microbiomes) analysis was carried out, as previously described [63].

To determine which of the study groups had a more diverse gut microbiota, and to indicate the differences regarding the genus and taxonomy levels of gut microbiota between patients with IBS and non-IBS control individual, the Kruskal–Wallis test with post-hoc multiple comparisons, via Dunn’s test, was applied. *p* ≤ 0.05 was defined as the level of statistical significance.

To determine which participants of the study had more diverse gut microbiota, four tests were performed: Bray–Curtis, Jaccard dissimilarity, and weight and unweight UniFrac using the QIIME 2 program, and the results were shown as the Principal Coordinates Analysis (PCoA).

The Kruskal–Wallis test was performed to assess the influence of (i) the IBS type (IBS-C, IBS-D, IBS-M, and IBS-U) on gut microbiota and (ii) gut microbiota on abdominal pain.

In these analyses, microorganisms that turned out to be important as a result of the differentiation of the studied groups were evaluated. Statistical analysis was performed to account for the variation observed in the studied groups, focusing on the selected microorganisms that were deemed relevant for differentiation.

## 5. Conclusions

There are differences in the microbiota composition between samples derived from patients with IBS and individuals without gastrointestinal symptoms. Patients with IBS were characterized as having a more diverse gut microbiota. When assessing the gut microbiota, the gender criterion should be taken into account.

Despite numerous studies about the gut microbiota, many aspects of the pathogenesis of IBS remain unexplained. Based on the presented data and previously published reports, it can be concluded that IBS should be considered a disease that requires a broad comprehensive approach in diagnostic aspects and a detailed individual analysis. In the future, knowledge about gut microbiota will allow the development of tools to improve the diagnosis and treatment of IBS.

## Figures and Tables

**Figure 1 ijms-24-10424-f001:**
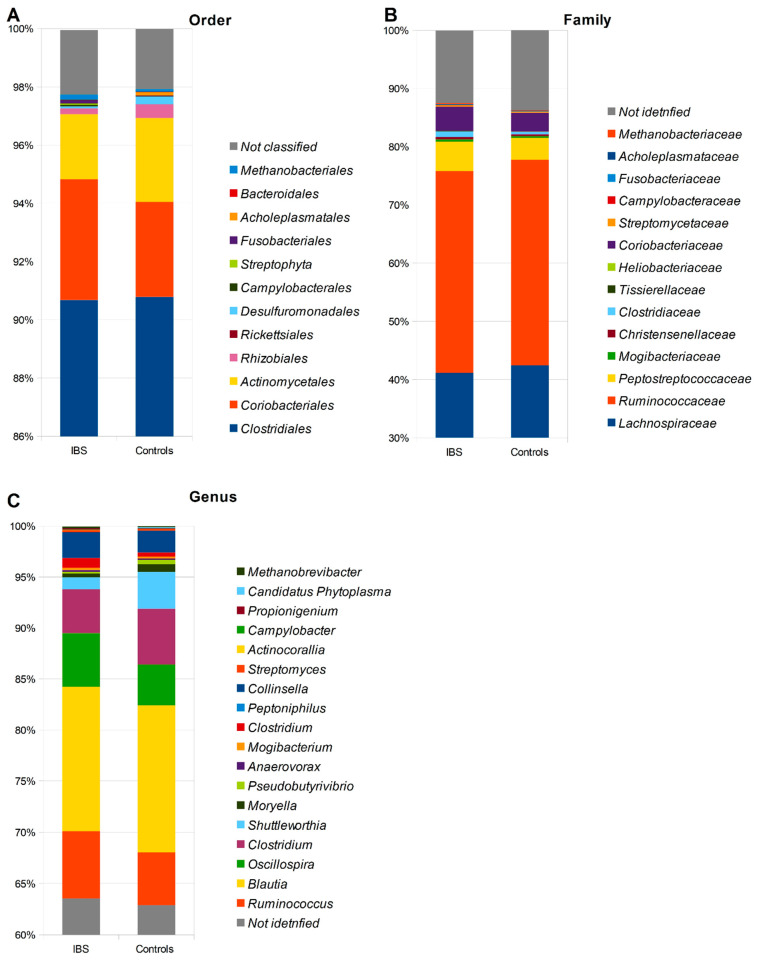
The relative abundance of microbiota in stool samples obtained from patients with IBS and non-IBS control group regarding the order (**A**), family (**B**), and genus taxonomy (**C**) levels.

**Figure 2 ijms-24-10424-f002:**
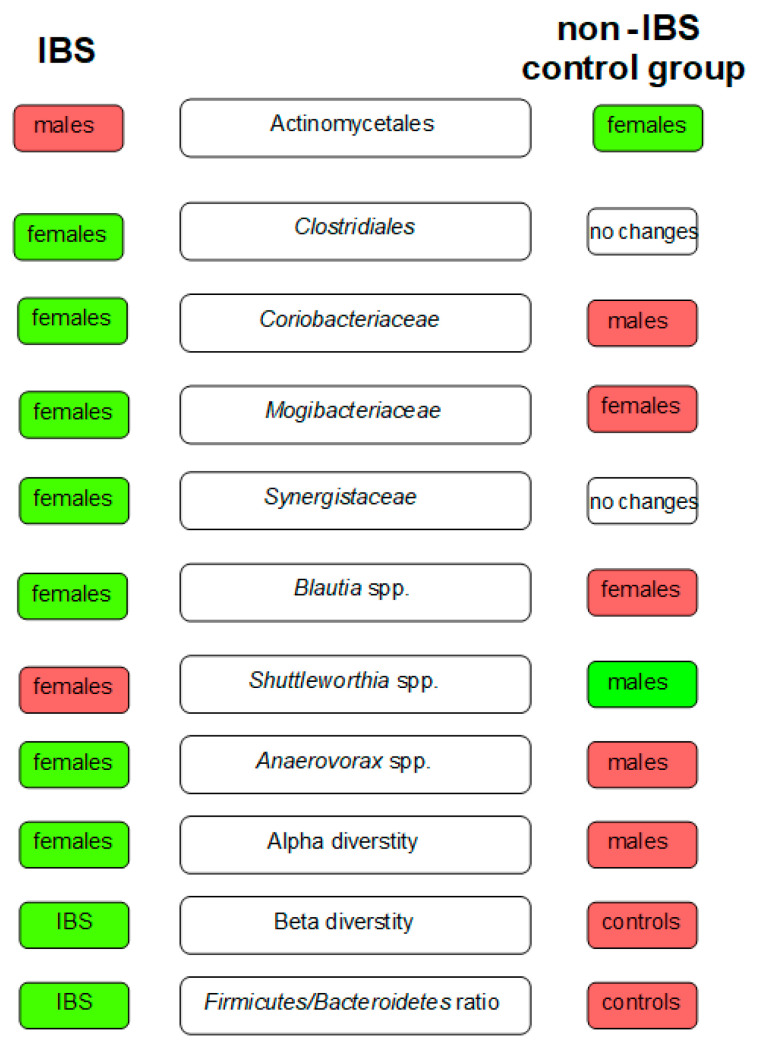
The key differences in gut microbiota between patients with IBS and the non-IBS control group. The green boxes indicate the microbiota of higher abundance among females/males in the study groups and the red ones of lower abundance.

**Figure 3 ijms-24-10424-f003:**
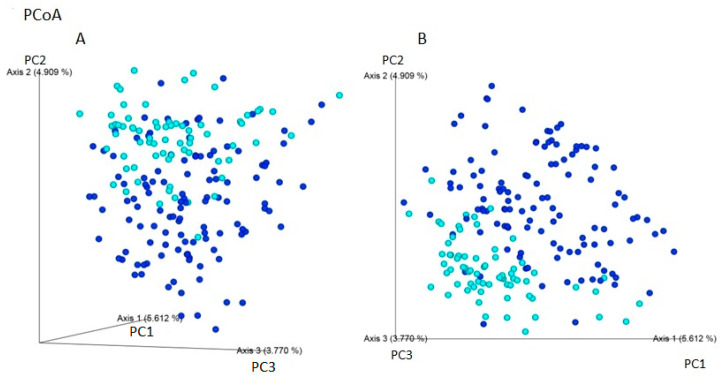
Beta-diversity assessed by the (**A**) Bray–Curtis and (**B**) Jaccard dissimilarity tests between individuals with IBS (navy blue) and non-IBS individuals—CI (light blue) based on ASVs (PCoA analysis). The gut microbiota of patients with IBS may be more variable in composition of microorganisms. The percentage of the total variance for each axis is shown in parentheses. The composition of microorganisms in patients with IBS and non-IBS control group is presented. The percentage of the total variance for each axis is shown in parentheses.

**Figure 4 ijms-24-10424-f004:**
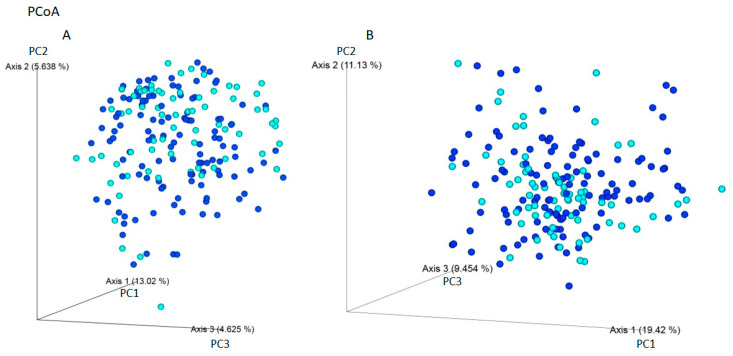
Beta-diversity assessed by the (**A**) unweight-UniFrac and (**B**) weight-UniFrac tests between IBS individuals (navy blue) and non-IBS individuals - CI (light blue) based on ASVs (PCoA analysis). The phylogenetic origin of the intestinal microbiota, taking into account the amount of ASVs, is similar among the studied groups. The percentage of the total variance for each axis is shown in parentheses.

**Table 1 ijms-24-10424-t001:** Characteristics of the study groups of patients with IBS and non-IBS control group, as well as the subgroups of patients with particular types of IBS, the IBS-C IBS-D, IBS-M, and IBS-U.

		IBS Type	Age (Years)			BMI *		
			$\bar{x}$	SD	Me	$\bar{x}$	SD	Me
IBS	female (N = 70)	IBS-C (N= 14)	41.6	13.9	38.8	23	2.4	23
		IBS-D (N = 24)	36.3	13.3	34	23.2	5	21.1
		IBS-U (N = 1)	62	-	-	24.2	-	-
		IBS-M (N = 16)	37.9	11	35.5	22.5	2.4	23
		not classified (N = 15)	40.5	10.5	38	22	2.2	21
	male (N = 51)	IBS-C (N = 10)	36.3	10.4	35.5	22.5	2	21.6
		IBS-D (N = 8)	34.2	12.9	29	25.5	7.3	24.8
		IBS-U (N = 7)	33.1	8.3	32.5	23.9	2.3	23.5
		IBS-M (N = 19)	30.4	8.1	29	24.1	3.2	25.3
		not classified (N = 7)	44.2	9.4	40	26.4	3	26.5
Non-IBS control group	female (N = 40)		32.21	10.91	30.5	24.25	5.04	24.3
	male (N = 30)		29.5	13.39	27	25.06	3.32	24.2

* BMI—Body Mass Index. Due to insufficient clinical information, 17 women and seven men were not classified into any of the IBS subgroups. $\bar{x}$—mean; Me—median; SD—standard deviation.

**Table 2 ijms-24-10424-t002:** Characteristics of patients with IBS regarding the questionnaire answers.

Questionnaire Questions	Patients with IBS	
	Females	Males
*How often do you defecate?*		
- several times a day	45%	42%
- once a day	37%	42%
- less often than once a day	18%	16%
*What is the form of your stool?*		
- diarrhoea	33%	39%
- constipation	26%	20%
- diarrhea/constipation (alternating)	9%	4%
- normal	32%	37%
*How often did you experience discomfort or pain in the lower abdomen in the last 3 months?*		
- every day	48%	33%
- more than once a week	35%	49%
- 2–3 days per month	10%	18%
- once a month	3.5%	0%
- less than once a month	3.5%	0%
*How often did this discomfort or pain decrease or disappear after a defecation*?		
- never	4%	10.5%
- sometimes	37%	37.5%
- often	31.5%	33%
- frequently	20%	12.5%
- always	7.5%	6.5%
*Were the bowel movements less frequent than the pain or discomfort?*		
- never	36%	44%
- sometimes	24.5%	27%
- often	24.5%	19%
- frequently	13%	4%
- always	4%	6%
*Have you had trouble sleeping since the pain started?*		
- yes	20%	27%
- no	80%	73%

## Data Availability

The data that support the findings of this study are available in the Appendix A of this article. Raw sequence reads are available from the corresponding author upon reasonable request. The BioProject (PRJNA981704) and associated SRA metadata are available at http://www.ncbi.nlm.nih.gov/bioproject/981704 (accessed on 10 June 2023).

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
