# Peer review of "Gender Influences Gut Microbiota among Patients with Irritable Bowel Syndrome"

_ijms, 2023, doi:10.3390/ijms241310424_

Round 1
Reviewer 1 Report
In the present original article Pecyna et al explored the composition of microbiota in patients with irritable bowel syndrome (IBS) compared to controls. Interestingly, they found that the composition may be influenced by gender. Main comments:
1) Exclusion criteria: what about of PPI consumption? They are known of modify microbiota composition.
2) In table 1 some important data have not been collected such as smoking habits, previous history of abdominal surgery or comorbidities (for example diabetes, which is strongly influenced by microbiota).
3) I think that PCA graph representing diversity should be embedded in the main text.
4) Line 119: this subheading is hard to understand.
5) Were Faecalibacterium prausnitzii and Akkermansia muciniphila analyzed?
6) I do not understand the relevance of the questionnaire in Table 1. Authors could have used a validated one for IBS such as GSRS score.
none
Reviewer 2 Report
The manuscript seeks to compare and characterize potential differences in microbiota between patients suffering from irritable bowel disease and healthy individuals, with a focus on eventual gender differences. As such the paper presents interesting results and contributes to the broader field insufficiently studied pertaining to Irritable Bowel Syndrom.
The subject is well within reach of the stated goals of the journal but minor corrections are advised to be made before publishing.
1) The classical IBS perturbation indicator showed as a telltale sign is the perturbation of the ratio between bacteria phyla especially between Firmicutes and Bacteroidetes yet this study completely overlooks the phylum taxa. We advise that at least a few sentences should be written about the situation at the phylum level. If the raw data was processed with Qiime2 this should not be a problem since the percentages were already presented in the S1 table. This does not change the quality of the data shown, it just completes the overall image. A graphical representation could as well be added to Figure 1.
2) In Figure S3 the total variance of the PCoA or the explanatory power of that analysis is very low with 5.612% being the Axis 1 value. I would advise against coming to a hard conclusion based on such low variance. Your conclusion was “patients with IBS were more variable in composition of microorganisms”, although there are many other papers stating that IBS actually lowers the variety. The difference can be seen very well in the dissimilarity PcoA as they form two separate clusters.
3) Please state in Figures S3 and S4 that the images are actually PCoA analyses representations even though it was stated in the article text. It can be misleading.
4) Line 226, it does not follow that a condition affecting humans should mirror the effects on mice and vice-versa. To stand to scrutiny it should be compared to a study done with human patience similar to this one, even though the conclusion is correct and is verified by other human studies.
5) The raw data obtained should be uploaded to an online repository with open access.
6) You claim in line 130 that "At the same time, other studies indicated that an intestinal dysbiosis is the cause of the disease, resulting from abnormalities in the quantitative and qualitative composition of the gut microbiota" but is the dysbiosis the cause or merely an effect? Correlation does not equal causation and this matter is far from being settled. Be advised to add a comment that reflects this point.
7) The authors are advised to involve a native speaker to help in editing the text. Parts of the manuscript are in need of more clarity/readability
Line 30 "comparing to do" should read compared to
Line 62 "are compiled" should read "were compiled"
Line 63 "and controls" should read "and as the non-IBS control group"
Line 69 " 10 types" ??? should read phyla
Line 130 "in patients appears who have" makes no sense. Please correct
Lines 187-190 long sentence. Consider breaking it into simpler sentences.
Line 222 "a gender" should be just gender
Line 238 "gender" should read genders
Line 249 "On opposite" makes little sense, please revise
Line 259 "it is proved" should read "It was proven"
In many cases, the wording is very difficult and articles are missing completely and commas are misplaced or missing entirely. Further editing is strongly advised.
Round 2
Reviewer 1 Report
answers are OK